# A Randomized Controlled Trial Evaluating the Relative Effectiveness of the Multiple Traffic Light and Nutri-Score Front of Package Nutrition Labels

**DOI:** 10.3390/nu11092236

**Published:** 2019-09-17

**Authors:** Eric A. Finkelstein, Felicia Jia Ler Ang, Brett Doble, Wei Han Melvin Wong, Rob M. van Dam

**Affiliations:** 1Program in Health Services and Systems Research, Duke-NUS Medical School, Singapore 807443, Singapore; felicia.ang@u.duke.nus.edu (F.J.L.A.); brett.doble@duke-nus.edu.sg (B.D.); melvin.wong@duke-nus.edu.sg (W.H.M.W.); 2Saw Swee Hock School of Public Health, National University of Singapore, Singapore 807443, Singapore; ephrmvd@nus.edu.sg

**Keywords:** front-of-pack labeling, nutrition labeling, food Intake, diet quality, diet, online grocery store, nutri-score, 5-color nutritional label, multiple traffic lights

## Abstract

The objective of this trial was to test two promising front-of-pack nutrition labels, 1) the United Kingdom’s Multiple Traffic Lights (MTL) label and 2) France’s Nutri-Score (NS), relative to a no-label control. We hypothesized that both labels would improve diet quality but NS would be more effective due to its greater simplicity. We tested this hypothesis via an online grocery store using a 3 × 3 crossover (within-person) design with 154 participants. Outcomes assessed via within person regression models include a modified Alternative Healthy Eating Index (AHEI)-2010 (primary), average Nutri-Score, calories purchased, and singular measures of diet quality of purchase orders. Results show that both labels significantly improve modified AHEI scores relative to Control but neither is statistically superior using this measure. NS performed statistically better than MTL and Control based on average Nutri-Score, yet, unlike MTL it did not statistically reduce calories or sugar from beverages. This suggest that NS may be preferred if the goal is to improve overall diet quality but, because calories are clearly displayed on the label, MTL may perform better if the goal is to reduce total energy intake.

## 1. Introduction

All nations have seen a significant upward trend in obesity rates over the past several decades, putting populations at increased risk for weight-related chronic diseases and premature mortality [1,2,3,4,5,6,7,8]. As a result, interventions aimed at encouraging healthier food consumption have been pursued by policy-makers worldwide. Singapore, the country of focus for this study, is no exception. In Singapore, 1 in 3 adults are overweight and 1 in 9 have diabetes [9]. These are substantial increases from previous decades [10]. To counter these trends, the government is considering several options. One policy under consideration is mandatory front-of-pack (FOP) labeling on nutrition content (www.reach.gov.sg). FOP labeling has been established as an effective tool to improve diet quality [11,12] and is now mandated in several countries including Chile and Ecuador [13], while numerous others, including Australia and New Zealand, have introduced voluntary labelling to complement the Nutrition Information Panel (NIP) appearing on the back of many products. 

Two labels not currently used but under consideration in Singapore are the United Kingdom’s Multiple Traffic Light (MTL) label (Figure 1, left panel) and France’s Nutri-Score (NS) label (Figure 1, right panel). For each food/beverage, the MTL separately presents major nutrient information with individual color-coded ratings for each nutrient based on reference intakes and the guidelines set out by the European Food Information Council. Contrarily, the French NS label presents a single summary score representing the overall diet quality for each food/beverage on a five-point color-coded scale from green (best) to red (worst) using the British Food Standards Agency nutrient profiling system.

Both labels have been shown to be effective against no label control conditions or against other FOP labels [14]. However, only hypothetical head-to-head studies that do not involve actual purchases have been conducted [15,16]. These studies, based on hypothetical purchases, suggest that NS may outperform MTL in promoting overall diet quality, possibly because of the high cognitive load required to understand the MTL and because seeing multiple attributes, with some good and some bad, may create decisional conflicts [17]. NS resolves both of these concerns by providing a single summary measure of the diet quality of the food/beverage. Therefore, although we hypothesize that both the MTL and NS labels will improve overall diet quality, as measured via a modified version of the Alternative Healthy Eating Index-2010 (AHEI-2010) or the weighted (by servings) average Nutri-Score of the shopping basket (with A = 5 and E = 1), relative to a no-label Control, we expect the effect will be greater for the NS label. AHEI-2010 is a validated index of diet quality and higher scores have been strongly associated with lower risk of major chronic diseases [7]. However, for singular measures of diet quality, including total and per serving values of: sugar (g), energy (kcal), fat (g), saturated fat (g), sodium (mg), fiber (g) and protein (g), we hypothesize that MTL will outperform NS as those who care specifically about these measures will see the values directly on the MTL label along with the percentage of daily recommended intakes. 

We also hypothesize that being hungry or in an unhappy mood at the time of shopping moderates the effectiveness of the labels. This is possible as negative mood and hunger have been associated with greater impulsivity [18]. Thus, shoppers with these attributes may be more likely to ignore the labels altogether. We also test for moderating effects of education and income. Those with greater education may be more likely to pay attention to and use the labels [19,20] and because healthier foods tend to be more expensive [21], income may also moderate the effect of any FOP label. As healthier foods tend to be more expensive, we also test whether either label increases total expenditure per shopping trip ($) and calorie per dollar (kcal/dollar). We test these hypotheses using an experimental online grocery store in Singapore where food is purchased and delivered to participants’ homes. 

## 2. Materials and Methods 

### 2.1. Design and Participants

Participants were recruited via Facebook and Instagram advertisements from September to November 2018. Prospective participants were directed from digital posters to the study website and asked to complete an online screener to determine eligibility. Prospective participants were offered the chance to participate if they were residing in Singapore, 21 years of age or above, and registered RedMart (a large on-line retailer in Singapore) shoppers. Recruiting existing online grocery shoppers ensured that participants would be familiar and comfortable with online shopping. 

Those interested and eligible were asked to complete: 1) a registration form containing name, mobile phone number, and email address; 2) an online consent form; and 3) after obtaining consent, a baseline demographics questionnaire. The study purpose and research hypotheses were not revealed to participants until their participation concluded, as this could have influenced their purchasing behavior. All participants received a debriefing summary with full details of the study upon study completion. Upon completion of all forms, the website created the participant’s account and unique participant identification number (PID) for use throughout the study. Participants then received an automated email with their unique login details and were asked to logon to the NUSMart online grocery store to complete the first of three shopping tasks. 

NUSMart is an online experimental grocery store developed by the study team and used to run the present trial (https://nusmart.duke-nus.edu.sg/NM). At the time of the trial, NUSMart contained over 4,000 products commonly purchased in local supermarkets (food and beverages only) in Singapore. Food and beverage items are sorted into various categories, such as dairy products, carbonated soft drinks, fresh meats and seafood, and snacks. Participants are able to add and remove products to and from their online grocery cart and review their cumulative total cart cost. The online grocery store was designed to mirror actual online grocery stores available in Singapore such as Fairprice Online (https://www.fairprice.com.sg/) and RedMart (https://redmart.lazada.sg/#home) in look and feel. All products include pictures of the item, retail prices and product descriptions. Nutrition Information Panels and product information are available on click-through.

Over the course of three weeks, participants logged on to the NUSMart website once a week and were asked to purchase their weekly groceries with a minimum spend of $50 and maximum spend of $100. Each participant therefore shopped a total of three times during the study, including one shop in each of three shopping conditions. A minimum and maximum spend ensured that participants completed a typical weekly grocery order and that no outliers would skew the data.

Participants shopped with the knowledge that, for each shop, they had no more than a 1 in 3 chance of being required to purchase the chosen foods using their credit card. The requirement to purchase would only be revealed upon spinning a digital “Wheel of Purchase” after hitting the checkout button that allowed for recording their weekly shop. This design was chosen to increase the likelihood that the purchases were an accurate reflection of the participants’ actual shopping patterns, lending credibility of the results over alternative designs that rely only on hypothetical shops. The grocery orders that needed to be fulfilled were repurchased by the study team and delivered via RedMart.

Upon completion of each order, participants completed a short post-shop survey to establish their mood and how hungry they were at the time of placing their order. This information was analyzed to test if these variables moderated the influence of the labels. Participants that completed all study elements were compensated with a $75 Lazada electronic gift voucher (a popular eCommerce website in Singapore, https://www.lazada.sg). 

The study protocol was approved by the Institutional Review Board, National University of Singapore (S-18-189) and registered on Clinicaltrials.gov under the number NCT03761342. All investigations were conducted according to the principles expressed in the Declaration of Helsinki and all participants provided written (electronically online) informed consent before being enrolled in the study. 

### 2.2. Interventions and Outcomes

The study was a crossover trial where all participants were exposed once to three shopping conditions in random order. Participants were randomly assigned to one of six intervention sequences via random permuted blocks of size three with equal allocation for the six sequences (see Appendix B
Table A8) by a computer program. Participants were blinded to intervention allocation, which was allocated via the NUSMart system. Allocation results were recorded within NUSMart and all investigators were blinded to group allocation. 

Arm 1 was a Control condition that mirrors a traditional web-grocery store with back-of-pack NIPs, but with no FOP labels. Arm 2 (MTL condition): is similar to Arm 1, with Multiple Traffic Light labels displayed on the FOP of all products. MTL was applied based on the NFP without modification of the original algorithm [22]. Arm 3 (NS condition) is similar to Arm 2, with Nutri-Score labels instead of MTL labels displayed on the FOP of all products. For the 3343 foods in the store, Nutri-Score was applied without modification of the original algorithm [23]. 26% (876 products) received an A grade, 12% (409 products) received a B grade, 26% (880 products) received a C grade, 25% (849 products) received a D grade, and 9.8% (329 products) received an E grade. For the 832 beverages, we applied a modified score based on the Singapore Health Promotion Board’s (HPB) proprietary scoring system that maintains a greater focus on calories and sugar content and aims to provide a greater distribution of scores compared to using NS without modification. Using NS without modification, 87% of drinks would have been assigned a D or E grade. Under the modified scoring: 29% that contained no sugar were assigned an A grade, 11% received a B grade, 3% received a C grade, 3% received a D grade, and 54% received an E grade. Appendix B
Table A9 shows the full list of beverages and their scores. Figure 2 shows an example of the NUSMart storefront with a sample of the MTL and NS labels as they appeared on the same fictional product in each condition. 

Prior to each shopping trip, a 60-second introductory video briefly explaining the MTL or NS labels was shown to participants in the corresponding condition. This sought to educate shoppers about how to read and understand each label, given that the local population has limited exposure to and education on the MTL and NS labeling schemes. The store was set up such that a participant could not shop until the videos were viewed.

All outcomes were calculated using the sales orders submitted by participants via the online system. The primary outcome is diet quality per shopping trip, as measured by a modified index of diet quality, the Alternative Healthy Eating Index (AHEI-2010). The AHEI-2010 is an updated measure of diet quality from the original Alternate Healthy Eating Index. It was constructed based on foods and nutrients predictive of chronic disease risk that incorporate current scientific evidence on diet and health; higher scores on the AHEI-2010 are strongly associated with a lower risk of major chronic diseases and cardiovascular mortality [7]. Eleven of 13 AHEI-2010 components, including vegetables, fruits, whole grains, sugar-sweetened beverages and fruit juice, nuts and legumes, red/processed meat, trans fat, long-chain fats and sodium are scored from 0 (worst) to 10 (best) were used for the present study. We dropped alcohol because it is not sold in our store and polyunsaturated fatty acids, as this information was not available. Therefore a maximum score of 90 (perfect diet quality) as compared to the usual maximum of 110 was employed. As we were calculating the scores on a weekly grocery purchase for the household, we divided the grocery purchase by the number of adult household members and by seven days to obtain the “per-day-per-person” consumption. Our secondary measure of diet quality, average Nutri-Score of the shopping basket, weighted by serving size, was calculated by applying A = 5 down to E = 1 for each food purchased. Other secondary outcomes included per serving and total values of calories, saturated fat, total fat, sodium, and sugar, and to quantify the effect of the labels, total spend and calories per dollar spent. 

### 2.3. Statistical Methods

A power calculation revealed that 140 participants were required to detect effect sizes of 0.30 or larger between arms. The calculation assumed a two-tailed test, a significance level of 0.05, power of 0.90, adjustment for 3 comparisons, and a cross-over design. 

Data is analyzed from an intention-to-treat approach that conforms to the Consolidated Standards of Reporting Trials (CONSORT) standards for reporting of randomized trials. The primary analysis employs the following first-differenced model:(1)$$\Delta Modified\;AHEI_{is}=\alpha +\beta_{NS}NS+\epsilon_{is}.$$

The first difference model exploits the repeated observations of individuals by differencing away, and thus controlling for, time invariant heterogeneity (e.g., age, health consciousness etc.) between individuals. In this specification, the dependent variable is the difference in modified AHEI scores in each treatment condition and the Control condition. The constant term, α, measures the difference in the Modified AHEI score for the MTL condition relative to the control shop. NS is a dummy variable that is set to one for shops where NS labels appeared on all products. $\epsilon_{is}$ is the error term. The subscripts are for each individual, i, and each shop, s. Each participant generates two observations (NS vs control and MTL vs control). This model is then estimated via ordinary least squares (OLS) with errors clustered at the individual level, so as to account for correlation between repeated shops for the same individuals. Secondary outcomes are analyzed using the same model. With the exception of the modified AHEI-2010, which is a composite measure of the entire basket, we also run the above models separately for foods and beverages. AHEI-2010 should not be run for food and beverages separately since the index scores are dependent on dietary components that span all major food and drink groups, and are not scored in isolation.

In line with our hypotheses, we conduct the following tests: α>0, Testing for whether the outcome is significantly greater in MTL than control; $\alpha +\beta_{NS}>0$, Testing for whether the outcome is significantly greater in NS than control; $\beta_{NS}>0$, Testing for whether the difference in outcome is greater for NS than for MTL.

To test the moderating effects of mood, hunger, income and level of education, we interacted moderators with the treatment dummy, resulting in the below model:(2)$$\Delta Modified\;AHEI_{is}=\alpha +\beta_{NS}NS+\beta_{M}Moderator+\beta_{int}Moderator*NS+\epsilon_{is},$$
where Moderator values greater than or equal to the median values for hunger and mood are considered hungry and happy respectively. $\beta_{M}>0$ tests for whether the moderators (i.e., hungry, happy, high income, high level of education) differentially influence the relationship between MTL and AHEI scores. $\beta_{M}+\beta_{int}>0$ tests this relationship for NS. Each moderator regression was run separately. All statistical analyses were performed using STATA version 15.

## 3. Results

### 3.1. Sample

Participant flow and corresponding sample sizes are presented in Figure 3. 168 participants were recruited to participate in the study. 145 completed all three shops. 14 did not place any order, 7 placed one and 2 placed two orders. Using the first difference model, we could analyze those with at least two purchases but not those with only one, resulting in an analysis sample of 147 participants. Table 1 presents the characteristics of this sample. The sample was largely of Chinese ethnicity (93.51%) and the mean age was 34.40 years (standard deviation (SD) = 6.88). The average body mass index (BMI) was 23.31 kg/m^2^ (SD = 4.07). The majority (68.83%) were female. The characteristics of participants who dropped out did not differ from the analysis sample. The completed CONSORT checklist can be accessed as Appendix A.

Table 2 presents the unadjusted values of the primary and secondary outcomes for the Control. Regression output testing for differences between conditions is reported in Table 3.

The mean modified AHEI-2010 score was 41.81 in Control. Consistent with our hypothesis, both labels showed statistically significant increases in modified AHEI-2010 scores (Table 3). The estimated increase was 1.09 in NS (*p* = 0.04) and 1.16 in MTL (*p* = 0.04) compared to Control. Contrary to expectations, the effect was not significantly different between labels. The mean Nutri-Score was 3.41. There was an increase in Nutri-Score by 0.33 points (*p* < 0.01) in NS compared to Control and, consistent with our hypothesis, by 0.31 points (*p* < 0.01) compared to the estimated effect of MTL. The difference in average Nutri-Score of 0.02 between MTL and control was not statistically significant (*p* = 0.06). 

There were no significant moderating effects of mood, hunger, education, or income on the modified AHEI-2010 or average Nutri-Scores, suggesting that these factors did not differentially influence the effect of labelling on diet quality for on-line shopping. The results can be found in Table A7 in the Appendix B.

### 3.2. Estimated Treatment Effect on Secondary Outcomes

In MTL, consistent with our hypothesis, calories decreased by 19.75 kcal/serving (*p* = 0.01), fat decreased by 1.03g/serving (*p* = 0.03) and protein decreased by 0.83g/serving (*p* = 0.01) compared to Control. In NS relative to Control, total saturated fats per order decreased by an estimated 29.29 g (*p* = 0.01). No other differences were statistically significant. These results can be found in Table A1 and Table A2 of the Appendix B.

### 3.3. Treatment Effect on Foods

For the analyses on food items only, there was a significant effect of the NS condition on average Nutri-Score (0.21, *p* = 0.02) and total saturated fats (–29.79g, *p* = 0.01) relative to control. MTL led to a significant decrease of calories per serving by 20.56kcal/serving (*p* = 0.02) and fats per serving by 1.02g/serving (*p* = 0.03) relative to control. No other differences were statistically significant. Results of the food-only analyses can be found in Table A3 and Table A4 in the Appendix B.

### 3.4. Treatment Effect on Beverages

For the analyses on beverages only, NS improves average Nutri-Score by 0.72 (*p* < 0.01), with NS having a significantly higher effect than MTL by 0.52 (*p* = 0.01). Similar to the total basket, relative to control, MTL reduced calories, fats and protein per serving by -15.48 kcal (*p* = 0.01), −0.55g (*p* = 0.03) and −0.76 g (p = 0.01), respectively. MTL also decreased total sugar purchased by −66.83 g (*p* = 0.03). Results of the drinks-only analyses can be found in Table A5 and Table A6 in the Appendix B.

## 4. Discussion

Front-of-pack labeling has been identified by the Singapore government as one of four promising strategies to tackle nutrition-related diseases. Our results show that both the Multiple Traffic Lights and Nutri-Score labeling scheme employed led to statistically significant improvements in diet quality relative to no-labelling based on mean modified AHEI-2010 scores and average Nutri-Score. In the Singapore Chinese population, a 10-point higher AHEI-2010 was associated with a 21% lower risk of coronary artery disease [24]. Thus, assuming a linear relationship between improvements in AHEI and reductions in coronary artery disease, the observed 1.16 point improvement, were it to be sustained on a population level, could be associated with as much as a 2.4% decrease in coronary artery disease, which is a relative large effect for a low-cost intervention. Testing the effects of the label on risk factors for non-communicable diseases should be an area of future research. Neither label was superior to each other in terms of modified AHEI-2010 scores, but NS performed better than MTL in terms of average Nutri-Score. This should not be surprising given that the average Nutri-Score most closely tracks what consumers saw on the NS label. 

These results provide support for implementation of either label if the goal is to improve overall diet quality as assessed via modified AHEI-2010 and average Nutri-Score. However, if the goal is to reduce intake of calories or improve intake of a specific nutrient, MTL may be preferred. Unlike NS, MTL showed statistically significant reductions in calories and fats per serving purchased relative to control in the full basket, and reductions in total sugar purchased for beverages. This result should not be surprising if we believe calories, sugar and fat are what consumers are most concerned about. MTL allows these nutrients to be seen directly on the label along with per serving values as a percentage of daily recommended values, whereas NS only shows a summary measure that masks potential benefits (i.e., decreases) in these individual nutrient domains. As a result, if the goal is to decrease calorie and sugar intake or rates of obesity, as opposed to improve overall diet quality, MTL may be preferred. Our results did not show MTL to be statistically different in these specific domains relative to NS, but this may be due to a lack of statistical power. 

Moderator analyses revealed no differential effects by mood, hunger, education or income on measures of diet quality. Although other studies have found mood and hunger to influence impulsivity and attentiveness when shopping for food [18,25,26,27,28], and these may influence the effects of FOP labels, this effect may be attenuated for on-line shopping where purchases and consumption and more disconnected. The lack of differential effects by education and income is encouraging as it suggests all shoppers may equally benefit from the labels. However, all moderating results should be viewed with caution given that the study may have been underpowered to identify these differences. 

This study had several strengths including its within-person randomized controlled trial design, a fully functional online grocery store with thousands of products to choose from, and actual delivery of orders for a subset of shops so that consumer shopping behavior was more likely to mimic an actual purchase. However, the study also has several potential limitations. Shopping was limited to one shop per condition and results may differ with repeated shops. The relative effectiveness of each label may also differ if the underlying algorithms used to define the labels were modified or if alternative summary measures of diet quality were used. AHEI is an accepted measure of diet quality at the individual level but is less frequently used to assess the quality of shopping baskets. Average Nutri-Score is less common but, as we show in Appendix B
Table A10, baskets that score better in this value also score better in AHEI and in each measure of diet quality listed on the NFP. However, as it is (by design) most strongly correlated with the NS label, it is biased in favor of this label. Future studies should consider alternative measures of overall diet quality. On-line shopping may also differ from in-store shopping, making it difficult to generalize the findings of this study to brick-and-mortar stores. Our sample is also not representative of the broader population. It includes a greater percentage of females, Chinese, and those with university education. Future studies should test these labels over repeated purchases and in different shopping venues, including on-line and in-person grocery and convenience stores, and with a broader subset of the population to explore whether results are sustained and generalizable and to identify which population subsets (e.g., dieters, more nutrition knowledgeable) are most likely to benefit from each type of label. Furthermore, showing participants the 60-second introductory videos may have primed them to shop for healthier products. However, given their lack of exposure to these labels we found it necessary to educate them on how to use the new labels as would occur with any effort to implement such labels in Singapore. As a final limitation, it is worth noting that our trial focused on the effects of FOP labelling on consumer choices, but effective labels could also encourage suppliers to reformulate (a positive outcome) and/or change prices in ways that may undermine some of the positive effects of the label. Real world studies are needed to explore these effects. 

## 5. Conclusions

Both the MTL and Nutri-Score front-of-pack labels improved dietary quality according to the modified AHEI-2010, providing support for implementation of either label if the goal is to improve overall diet quality. NS outperforms MTL and no FOP labels in average Nutri-Score but, unlike MTL, does not reduce calories, suggesting that MTL may be preferred to NS if the goal is to reduce caloric intake and obesity rates. 

## Figures and Tables

**Figure 1 nutrients-11-02236-f001:**
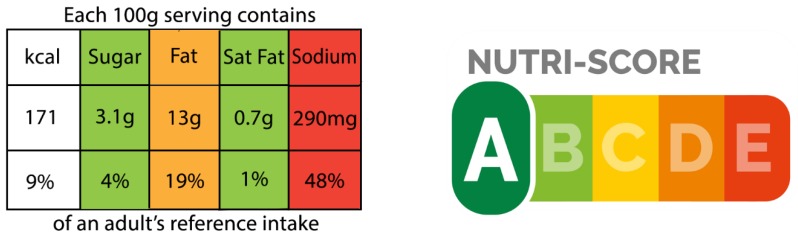
Front-of-pack labels under consideration in Singapore.

**Figure 2 nutrients-11-02236-f002:**
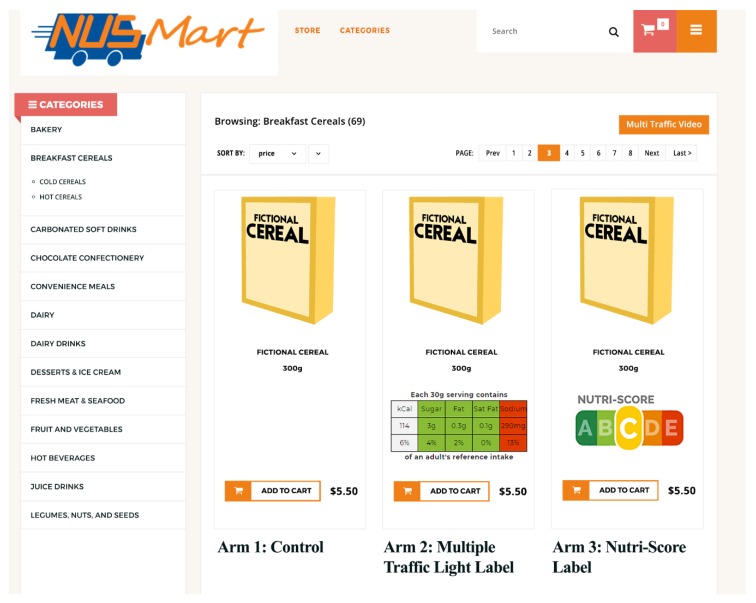
Example of the storefront with a sample of the Multiple Traffic Light and Nutri-Score labels on the same product as it appears in each condition.

**Figure 3 nutrients-11-02236-f003:**
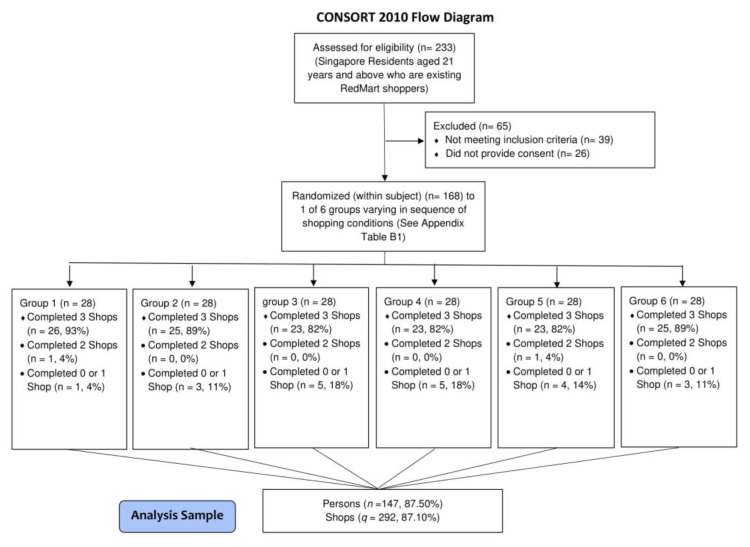
CONSORT flow diagram for participant recruitment and randomization.

**Table 1 nutrients-11-02236-t001:** Descriptive statistics.

Variable	Analysis Sample (*n* = 147)	Drop-Out Sample (*n* = 21)
	Mean/% (S.D.)	Mean/% (S.D.)
Age (years)	34.69 (6.83)	33.17 (8.12)
Body mass index (BMI) (kg/m^2^)	23.31 (4.07)	24.06 (4.96)
Female (%)	68.71 (5.62)	83.33 (4.52)
Ethnicity (% Chinese)	93.20 (3.05)	100 (0)
University education and above (%)	67.01 (5.70)	66.67 (5.72)
Household income $10,000/month and above (%)	32.65 (5.69)	12.50 (4.01)

**Table 2 nutrients-11-02236-t002:** Dietary characteristics of food purchases in the Control condition (*N* = 147).

Outcome	Unadjusted Mean	95% CI
Modified Alternative Healthy Eating Index (AHEI-2010) Score	41.81	(40.71, 42.92)
Average Nutri-Score	3.41	(3.33. 3.50)
Total energy (kcals) (in 1000s)	9.72	(8.40, 11.04)
Total sugar (g) (in 1000s)	2.68	(2.25, 3.10)
Total fat (g) (in 1000s)	4.41	(3.38, 5.43)
Total saturated fat (g) (in 1000s)	1.57	(1.02, 2.12)
Total sodium (mg) (in 1000s)	108.74	(80.94, 136.53)
Total fiber (g) (in 1000s)	0.92	(0.73, 1.11)
Total protein (g) (in 1000s)	3.16	(2.59, 3.73)
Mean calories per serving (kcal/serving)	127.74	(117.60, 137.87)
Mean sugar per serving (g/serving)	4.57	(3.80, 5.35)
Mean fat per serving (g/serving)	5.21	(4.59, 5.83)
Mean saturated fat per serving (g/serving)	1.83	(1.58, 2.08)
Mean sodium per serving (mg/serving)	144.49	(116.95, 172.03)
Mean fiber per serving (g/serving)	1.19	(1.01, 1.37)
Mean protein per serving (g/serving)	4.66	(4.03, 5.28)
Total spend ($)	53.86	(52.41, 55.31)
Calorie per dollar (kcal/$)	249.63	(231.20, 268.05)

**Table 3 nutrients-11-02236-t003:** Estimated effects of Multiple Traffic Lights (MTL) and Nutri-Score labels (*n* = 147, *N* = 292).

Outcome	AHEI-2010	Average Nutri-Score
α (MTL vs Control)	1.16 *	0.02
(s.e.)	(0.53)	(0.08)
$\beta_{NS}\;$ (incremental effect of NS over MTL)	−0.07	0.31 **
(s.e.)	(0.58)	(0.09)
($\alpha +\;\beta_{A})$ (NS vs Control)	1.09*	0.33 **
(s.e.)	(0.53)	(0.09)

* *p* < 0.05, ** *p* < 0.01.

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
