# Peer review of "A Randomized Controlled Trial Evaluating the Relative Effectiveness of the Multiple Traffic Light and Nutri-Score Front of Package Nutrition Labels"

_nutrients, 2019, doi:10.3390/nu11092236_

Round 1

Reviewer 1 Report

In this manuscript, the authors compare the effects of two systems of nutrition labeling on the nutritional quality of purchases made through an online experiment.  

I liked to topic of the manuscript as nutrition labeling and its effect on choice and behavior is a central aspect of food policy. The manuscript flows very well and is generally well-written.

However, in order to improve the manuscript, I would recommend the following :

-          Introduction : it would be helpful to provide key figures about the health situation in Singapore such as obesity rate for example.

-          Sample composition and size : I am not very familiar with the Singaporian demographics however highly-educated consumers and female seem to be over-represented in the sample. This is probably due to the fact that participants have been recruited through Facebook and Instagram. This should be taken into account by the authors as gender and education level are important determinants of nutrition information use (Nayga, 1996). Considering this, the authors should interpret with caution their findings. In addition, some unsignificant effects could be due to the fact that the study is underpowered (as rightly noticed by the authors).

-          Secondary hypothesis : I appreciate that the authors have included individual variable measurements in their questionnaire but given that the research objective is to compare MCL and NS, it would have been more relevant to examine variables that could have lead participants to prefer one to the other. For instance, MCL could have a bigger impact on dieters (vs. non-dieters) as they are motivated to reduce their calorie intake (Trudel et al., 2015). Similarly, nutrition knowledge could influence consumers’ use of NS as it is supposedly easier to understand (Ducrot et al., 2015).

-          Method description : the notion of cross-over trials is not clear to me. Does this mean that this is a within-subjects design ? In Figure 3, for each group, it would be helpful to add the type of nutrition labels displayed to participants (MCL ; NS ; control).

-          I have also another comment about label familiarity. What was the initial level of familiarity with each label. Some food companies may already use one of this label in Singapore.

-          Additional analysis : generally, nutrition labels do not influence choice and behavior equally among food categories (Ducrot et al., 2016). Did you include food categories as a moderator in the model tested ? Gender should also be added as a potential moderator as female tend to pay more attention to calorie information than male, which is only available in the MCL (Krieger et al., 2013).

As I mentioned, I liked the topic of the authors' manuscript and I thought that their idea was generally well communicated throughout the studies. I hope the authors find my feedback useful and I wish them all the best with their research!

Reviewer 2 Report

This study experimentally examines the effectiveness of two FOP labelling systems relative to a no-label control on the healthfulness of consumer food and beverage purchases in an online supermarket shop. Generally, I think this study is promising, is well designed, and will make a significant contribution to the evidence base. However, before considering it for publication, I suggest strengthening the introduction, removing the Nutri-Score outcome entirely, and discussing the results of the secondary outcomes and how these results fit with the current evidence. Specific recommendations are listed by section below.

Abstract

You use diet quality and nutritional quality interchangeably in the abstract and throughout the paper. Can these terms be used interchangeably? Be clear that the RCT tests the two FOP labelling systems relative to a no-label control. List acronym for AHEI beside first mention. Lines 11-12, consider: The objective of this trial was to test two promising front-of-pack nutrition labelling systems, the UK’s Multiple Traffic Lights (MTL) and 2) France’s Nutri-Score (NS), relative to a no-label control. Is this study design a cross-over design or a within-participants design? What is the Nutri-Score outcome? If it’s simply assessing purchases by letter ratings, this is biased towards the one labelling system. I strongly suggest removing the Nutri-Score outcome from the study entirely, as mentioned in my comments above. Reconsidering your conclusions based on the results for the AHEI outcome and secondary outcome.

Introduction

In the first paragraph: please be more specific about how obesity rates have significantly increased over the 30 years (as rates in many high income countries are now plateauing) FOP labelling systems have been established as effective tools for supporting consumer awareness and understanding of nutrition information, and to some degree, supporting healthier food purchases. To my knowledge, some studies have examined the impact of FOP labelling systems on consumer food purchases in supermarkets, but none have investigated diet quality? Although countries have mandated or voluntary FOP labelling systems, these systems vary across countries in terms of format and the nutrient profiling system underpinning the label. This needs to be addressed. Second and third paragraphs: A more thorough and comprehensive discussion of the MTL and NS systems are required, specifically the key distinguishing features (e.g., nutrient-based vs summary system, calories included or not, both include colour coding, MTL is the most well studied and established FOP system, NS is relatively newer). Line 45: How have these two FOP systems outperformed other systems (e.g., consumer comprehension, use; or on purchases)? Why might these two systems be superior to other FOP labelling systems tested in previous studies? Which types of FOP systems have they been outperformed?

Materials and Methods

Overall, this section is strong. However, I did find myself asking questions then later finding the information further down in the methods. Line 76: Why were participants 21 years of age and above? Lines 79-81: What were participants told about the study during the consent process? Were they told the purpose of the study? If yes, how would this information influence participants’ purchases? Lines 94-95: In FOP label conditions, did participants have to click through to see the FOP label information or was the FOP label information below the product image from the start? Additionally, more details are required about where the FOP label information is displayed on-screen and the size of this information. Plus, how many products are shown on-screen at one time? Is it possible for participants to easily compare across products on-screen? Why did you choose to expose participants to all 3 conditions instead of assigning each to the same 1 condition for 3 shops? What are the advantages/disadvantages of this within-subject approach? Be sure to use web-based and/or online grocery store consistently throughout the paper. I suggest avoiding the term web-grocery store (as used in line 126). Lines 133-141 – Is the modification made to rate beverages using the NS system consistent with what is actually being considered by the Singapore government? If not, I think the lack of variation in NS scores for beverages is a major limitation of the NS system and should be tested in the study. Does the 60-second introductory video prime participants to shop for healthier products regardless of the labels? Was a video explaining NIP’s shown before the no-label control shops? Please consider the influence of showing these videos to participants before going shopping. Lines 165-167: More information about the secondary outcomes is required. Are these outcomes assessed per serving, overall?

Results

I found Table 1 difficult to read as the rows were not easy to follow across the table. I would move the results for the secondary outcomes to the main paper, these outcomes are very interesting and need to be further highlighted throughout the paper. Keep the appendices to information and data that is not essential to understanding the results.

Discussion

Overall, the discussion will need to be rewritten without reference to the Nutri-Score outcome, and the conclusions about the two FOP label systems and which system’s performance is more superior, may also change. Is a difference of 1.09 and 1.16 in AHEI-2010, clinically meaningful? What exactly does it mean? A more thorough discussion of the results of the secondary outcomes is key and required. These findings are likely as valuable as the main finding.
